# Appropriate Imaging Modality for the Etiologic Diagnosis of Congenital Single-Sided Deafness in Children

**DOI:** 10.3390/jcm7120515

**Published:** 2018-12-04

**Authors:** Sang-Yeon Lee, Shin Hye Kim, Yun Jung Bae, Eun Hee Kim, Ja-Won Koo, Byung Yoon Choi

**Affiliations:** 1Department of Otorhinolaryngology-Head and Neck Surgery, Seoul National University Bundang Hospital, 300 Gumi-dong, Bundang-gu, Seongnam, Korea; maru4843@hanmail.net (S.-Y.L.); jwkoo99@snu.ac.kr (J.-W.K.); 2Department of Otorhinolaryngology-Head and Neck Surgery, Inje University Haeundae Paik Hospital, 875 Haeun-daero, Haeundae-gu, Busan, Korea; corn20_kr@hanmail.net; 3Department of Radiology, Seoul National University Bundang Hospital, 300 Gumi-dong, Bundang-gu, Seongnam, Korea; bae729@gmail.com; 4Department of Radiology, National Medical Center 245 Eulji-ro, Jung-gu, Seoul, Korea; kimeunheekeh@gmail.com

**Keywords:** single-sided deafness, cochlear nerve deficiency, magnetic resonance imaging

## Abstract

We aimed to compare the diagnostic yield between temporal bone computed tomography (TBCT) and internal auditory canal MRI (IAC MRI) for the etiologic diagnosis of children with congenital single-sided deafness (SSD) and the evaluation of cochlear implant (CI) candidacy. In the original cohort, 24 subjects with congenital SSD were enrolled and underwent both TBCT and IAC MRI. We recruited an additional 22 consecutive infants with congenital SSD (the supplementary cohort) and evaluated in particular the cochlear nerve (CN) integrity using IAC MRI. Cochlear nerve deficiency (CND) was classified as ‘absent’, ‘small’, and ‘indeterminate’ via mutual comparison between optical and parameters based on the MRI results. The most common etiologies were CND in the original cohort (19 out of 24). Notably, accurate evaluations of CN status (‘small CN’ = 2, ‘indeterminate CN’ = 2), inner ear malformations, and brain abnormalities were possible only with MRI. The ‘indeterminate CN’ tended to be more frequently detected in SSD ears than in unaffected ears. MRI appeared to be more accurate than TBCT in a meticulous differentiation of CN, which is crucial for the selection of appropriate CI candidacy among congenital SSD children. Additionally, we introduced the novel concept of ‘indeterminate CN’, of which the causal relationship with SSD awaits confirmation.

## 1. Introduction

Single-sided deafness (SSD), known as a form of profound unilateral sensorineural hearing loss (USNHL), is a specific condition characterized by the total loss of functional hearing ability (>90 dB) in the poorer ear and pure-tone average hearing threshold of ≤20 dB HL to 4 kHz inclusively in the better ear [1]. The resultant monaural auditory deprivation can lead to difficulties in speech and language development, as well as in educational and psychological development in children [2]. Unlike conductive hearing loss, in which the etiology in most cases is readily determined, approximately up to 60% of USNHL cases have an unknown etiology, requiring more refined and extensive investigation for the initial evaluation such as imaging, serologic testing, and genetic testing [3,4]. Early identification and etiology-specific rehabilitation are important factors to improve outcomes. Radiological evaluation for USNHL may be an especially efficient step because anatomical abnormalities in the inner ear and cochlear nerve are common, accounting for approximately 35% of all USNHL cases; based on imaging studies, the incidence of anatomical abnormalities seems to be dependent on the severity of hearing loss [5].

A substantial improvement after cochlear implantation (CI) was reported not only in children with acquired SSD [6] but also in limited early implanted (before 18 months old) prelingual congenital SSD cases with an intact cochlear nerve (CN) [7,8,9]. Early implantation within a ‘critical’ period for a favorable outcome from congenital or prelingual SSD cases might prevent aural preference with a reorganization of auditory pathway, leading to binaural development [7,8,9]. A cochlear nerve deficiency (CND) that coincides with a narrow bony cochlear nerve canal (narrow BCNC) is a frequently encountered anatomical abnormality accounting for the etiology of USNHL/SSD [10]. Notably, CI outcomes and candidacy appear to be largely dependent on CN integrity [11]. Thus, thorough and robust radiological evaluation of CN integrity within the critical period is particularly important.

High-resolution temporal bone computed tomography (TBCT) and internal auditory canal magnetic resonance image (IAC MRI) have been regarded as reliable imaging modalities to diagnose USNHL. They have been known to complement each other, due to their capacities in providing detailed images of the inner ear and CN. However, to date, there has been a lack of consensus whether CI candidacy can be confirmed using either TBCT or MRI, or both, especially for pediatric subjects with SSD [12,13]. Given this, this study aimed to compare the diagnostic yield between TBCT and IAC MRI, with special attention to CN integrity that affects CI candidacy. In addition, we explored the implications of meticulous CN evaluation in children with congenital SSD.

## 2. Materials and Methods

### 2.1. Subjects

Pediatric subjects with SSD between March 2012 and December 2015 were enrolled. Subjects were included if they showed a pure-tone average of 500, 1000, 2000, and 3000 Hz greater than 90 decibels hearing level (dB HL) in the affected ear in conjunction with pure-tone average at the same frequencies lower than 20 dB HL in the non-affected ear or no response to the auditory brainstem response (ABR), in case of absence of the audiometry test. All participants underwent a newborn hearing screening, including otoacoustic emissions and cochlear microphonics. Subjects with conductive components were excluded by bone conduction ABR.

After meticulous reviews of medical history and laboratory test, subjects with definite congenital SSD that were confirmed by the newborn hearing screening were included. Subjects were excluded from the analysis if they had no definite evidence of congenital onset. No subject in the congenital SSD group had a history of sudden hearing loss or etiologies, such as meningitis, head injury, brain surgery, neurological disorders, and cytomegalovirus infection. In addition, we evaluated the presence of the pigmentary disorder in our cohort with congenital SSD. In cases of suspecting a pigmentary disorder involving premature gray hair and freckles, we screened genes related to Waardenburg syndromes such as *PAX3*, *SOX2*, and *MITF* and excluded them from our cohort [1]. Eventually, 24 eligible SSD pediatric subjects (the original SSD cohort) were enrolled and underwent meticulous radiological evaluations by both TBCT and IAC MRI. The median age at imaging evaluation of SSD was 56 months (3–110 month), and there was no gender (13 males and 11 females) or side predominance (12 left and 12 right ears) among the subjects. 

Notably, from January 2016, we changed the initial imaging protocol to include only IAC MRI evaluation for pediatric congenital SSD subjects aged <5 years, after observing interim results suggesting a superior diagnostic yield of IAC MRI from our previous 24 SSD cohorts. Moreover, young age is known to be a risk factor for increased radiation hazards, which is the highest in children aged five years or younger. Even if the effective dose of radiation from TBCT has been classified as low, the biological effects of low-dose radiation can also lead to cancer development [14]. In this regard, from January 2016 to December 2017, we prospectively recruited 22 additional consecutive infants under the age of five with congenital SSD (the supplementary SSD cohort) and evaluated them using only IAC MRI, to focus on the characteristics of CND as the underlying radiologic etiology in SSD pediatric subjects. Accordingly, 46 pediatric subjects with congenital SSD were enrolled. The same imaging techniques and slice thickness were performed for both the original and supplementary SSD cohort. 

This study was conducted in accordance with the Declaration of Helsinki (IRB-B-1608-357-103). 

### 2.2. Analyses of Inner Ear Structures by TBCT

For TBCT examination, the axial and coronal images were obtained with 0.63mm slice thickness using 120 kV, 20 × 0.625 mm collimation, a 0.5-sec rotation, pitch factor of 0.248, and 100 to 200 mAs, in accordance with the age of subjects on 256-channel multi-detector computed tomography (Brilliance; Philips Medical Systems, Best, The Netherlands). Multi-planar coronal reconstruction from TBCT data of axial scan was additionally performed. The width of BCNC was measured at the mid-portion between the base of the modiolus of cochlea and the inner margin of the fundus of IAC, and the diameter of IAC was determined by drawing a minor axis perpendicular to the major axis of IAC on the porus level of axial plane via a software on the Picture Archiving and Communication System (PACS) workstation (Infinitt, Seoul, South Korea) (Appendix A). Narrow BCNC was defined as a width of <1.4 mm [15], and narrow IAC was defined as a width of <3 mm [16]. Anatomical abnormalities via TBCT were reviewed based on the structural specificities including diameter of BCNC, diameter of IAC, and inner ear (cochlea, vestibule, endolymphatic duct, semicircular canals).

### 2.3. Size Measurement of Inner Ear Structures in IAC MRI

IAC imaging was performed by a 3T MRI scanner (Achieva and Ingenia; Philips, Best, Netherlands), equipped with a SENSE head coil (Philips Healthcare) for signal reception. On the basis of T2-weighted turbo spin-echo acquisition (TR, the repetition time/TE, the echo time: 7395/120 ms; flip angle: 90°; slice thickness: 0.7 mm; no slice gap; field of view (FOV): 100 × 100 mm^2^; voxel size: 0.4 × 0.4 × 0.4 mm^3^), the oblique sagittal images that directly showed the presence of nerves in the IAC were reformatted in planes perpendicular to the course of the nerve in the midpoint of IAC of axial images (Figure 1A). On the oblique parasagittal image of the unaffected ear, the facial nerve (FN) lies superiorly, with the cochlear nerve (CN) inferior to it, within the anterior aspect of the canal. The superior (SVN) and inferior vestibular nerve (IVN) lie posteriorly (Figure 1B). The lengths of the major axis (long diameter, Ld) and the minor axis, which is perpendicular to the major axis (short diameter, Sd) of CN, on the oblique parasagittal images of both the unaffected and affected ears were measured to the nearest 0.1 mm by two neuro-radiologists blinded to all subjects’ information using a software on the PACS workstation combined with sharpening and zooming (Infinitt, Seoul, South Korea). The cross-sectional area (CSA) of CN was also measured as elliptical regions of interest (ROI) on the basis of Sd and Ld (Figure 1C).

The mean of Ld, Sd, and CSA in the unaffected ears from the present study were 1.19 ± 0.13 mm, 0.95 ± 0.16 mm, and 0.89 ± 0.18 mm^2^, respectively (Figure 2).

The CN status was classified in accordance with the presence or the size of CN: (1) ‘absent CN’ refers to the a parasagittal oblique image with no visible CN; (2) ‘small CN’ is defined as when CN of the affected ear is optically smaller than FN and the values of Ld, Sd, and CSA were all less than those corresponding to one standard deviation from the mean of the unaffected ear (Ld < 1.06 mm; Sd < 0.79 mm; CSA < 0.71 mm^2^); (3) ‘indeterminate CN’ is defined as when CN is optically similar to the FN in size but meets all diagnostic criteria values for ‘small CN’. In this study, CND included ‘absent CN’, ‘small CN’, and ‘indeterminate CN’ as defined according to their criteria described above. Based on IAC MRI, anatomical abnormalities were reviewed with regard to structural specificities, including CN integrity, inner ear (cochlea, vestibule, endolymphatic duct, semicircular canals), and brain lesions.

### 2.4. Statistical Analysis

All statistical analyses were performed using IBM SPSS Statistics (SPSS 20.0 K, IBM; Seoul, Korea), Korean version 20.0 for Windows. The Wilcoxon signed-rank test and Mann-Whitney *U* test were used to compare the variables between the affected and non-affected ears. The ability to detect CND between two imaging modalities was compared by Fischer’s exact test and Chi-square test. Inter-rater reliability was performed to determine the consistency among the rates via *Kappa* statistic. The consistency level was defined as follows: *Kappa* < 0.4, poor consistency; 0.4 ≤ *Kappa* < 0.75, general consistency; and *Kappa* ≥ 0.75, good consistency [17]. The criterion for statistical significance was set at *p* < 0.05.

## 3. Results

### 3.1. TBCT Findings from the Original SSD Cohorts

The two most commonly observed anatomical abnormalities in SSD ear, based on TBCT findings, were narrow BCNC (15 of 24) and narrow IAC (10 of 24), according to our criteria (<1.4 mm and <3 mm, respectively). The mean diameter of BCNC, on the axial image, in the affected ear was significantly narrower than in the unaffected ear (1.25 ± 0.73 mm vs. 1.89 ± 0.30 mm, *p* = 0.006); the mean diameter of IAC, however, in the affected ear was not significantly narrower than in the unaffected ear (3.77 ± 1.12 mm vs. 4.22 ± 0.83 mm, *p* = 0.096). Notably, none of the unaffected ears from all subjects showed a narrow BCNC or a narrow IAC on TBCT images. Among the 10 subjects with narrow IAC, eight had a narrow BCND and the other two had a normal BCNC. The lateral semicircular canal (LSCC) dysplasia was identified in one subject who showed a normal IAC and a narrow BCNC. Each incomplete partition type I (*n* = 1) and incomplete partition type II (*n* = 1) was identified among seven subjects who had a normal IAC and normal BCNC (Appendix A).

### 3.2. IAC MRI Findings from the Original SSD Cohort

There were good levels of inter-observer agreement for all measured variables in the unaffected ears (*Kappa* = 0.982, 0.968, and 0.932, respectively, all *p* < 0.001) (Table 1). 

According to IAC MRI, CND was the most commonly observed neuroanatomical characteristic in the affected ear, which was evident in 19 out of the 24 SSD subjects. Of these 19 subjects with CND, 15 were defined as ‘absent CN’ and four were defined as either ‘small CN’ (*n* = 2) or ‘indeterminate CN’ (*n* = 2) based on our criteria (Figure 3).

Among the 19 subjects with CND, 13 had isolated CND. Coexisting inner ear abnormality along with CND, such as vestibular nerve deficiency (*n* = 4), incomplete partition type I (*n* = 1), incomplete partition type II (*n* = 1), and lateral semicircular canal dysplasia (*n* = 1), were identified by IAC MRI (Appendix A). Interestingly, two subjects with incomplete partition (type I, *n* = 1; type II, *n* = 1) had a ‘small CN’ based on IAC MRI, whereas these same subjects showed normal IAC and normal BCNC based on TBCT. Moreover, one SSD case accompanied by an absent CN turned out to manifest multiple white matter lesions with CMV evidence.

### 3.3. Predictive Ability of the Status of the Vestibulocochlear Nerve based on TBCT Findings

In the original SSD cohort, the prediction of CND according to the IAC diameter (<3 mm) and BCNC diameter (<1.4 mm) were demonstrated in Table 2.

Among the original cohort, 15 subjects who showed a narrow BCNC (<1.4 mm) on the TBCT turned out to have an absent CN according to IAC MRI. On the other hand, four out of nine subjects with a normal BCNC, as estimated by the TBCT, turned out to have a milder degree of CND (‘small CN’ = 2, ‘indeterminate CN’ = 2) according to the oblique parasagittal MRI. It provides a significantly superior CND identification rate compared with estimating the size of the IAC or BCNC using TBCT *(p* = 0.003, *p* = 0.0456, respectively) (Table 3).

### 3.4. Cochlear Nerve Deficiency from the Original and Supplementary SSD Cohort

We prospectively recruited a supplementary SSD cohort that was evaluated only by IAC MRI to identify if the similar incidence of the ‘indeterminate CN’ obtained from our original cohort is replicable. With further analysis regarding CND in 22 subjects under the age of five, there were three types of radiologic etiologies, ‘absent CN (*n* = 11)’, ‘small CN (*n* = 2)’, and ‘indeterminate CN (*n* = 2)’ based on our proposed criteria. The incidence of CND (15 of 22, 68.2%) in the supplement SSD cohort was in line with that in the original SSD cohorts (19 of 24, 79.2%). Remarkably, ‘indeterminate CN’ was detected in two out of 22 SSD ears, while there was no ‘indeterminate CN’ in the unaffected ear. Collectively, ‘indeterminate CN’ was present in four ears out of 46 SSD ears from the original and supplementary cohort, while it was not present in the 46 unaffected ears. Furthermore, there were good levels of inter-rater reliability between two neuro-radiologists blinded to all subjects’ information for measured ‘small CN’ and ‘indeterminate CN’ variables in affected ears (*Kappa* = 0.983, 0.970, and 0.986, respectively, all *p* < 0.001). Taken together, in our entire cohort, ‘indeterminate CN’ appeared to be more frequently detected in SSD than in unaffected ears, which neared a statistical significance (4/46 vs. 0/46, Univariate Fischer’s exact test, *p* = 0.058). 

## 4. Discussion

Even though there are papers addressing the diagnostic yield between TBCT and MRI for CI candidates [18,19,20] they mostly focused on bilateral SNHL. Our manuscript merits special attention since we focused on SSD, not bilateral SNHL. Bilateral SNHL can be caused by significantly heterogenous etiology than does SSD. Therefore, it is conceivable that TBCT and MRI should complement each other to deal with a wide range of these inner ear abnormalities. Congenital SSD shows a rather homogenous etiology, which is mostly related to CN status. Since the etiology is limited, there may be less need for several imaging tests for diagnosis. In this study, MRI appeared to be advantageous over TBCT in a meticulous evaluation of CN which is crucial for selection of appropriate CI candidacy among congenital SSD children. Instead, in terms of CI candidacy, SSD requires a much higher standard and criteria especially about the CN status, which in turn would mandates higher resolution images specialized for CN status. For this purpose, we came up with the novel objective measure of CN integrity by MRI that could potentially reflect population of spiral ganglion cells. Furthermore, the MRI used in our study was superior to that in previous studies in terms of resolution and sequencing strategy.

### 4.1. Diagnostic Yield of Cochlear Nerve Integrity between TBCT and IAC MRI

CND has become a significant etiology of SSD with the advancement of MRI [10]. Despite the relatively small number of participants in this study, this is in line with recent research that reported CND was more common in younger children, especially in infants, and in those with severe-to-profound single-sided SNHL [21]. A new sequencing strategy and resolution advancement of MRI allows for an accurate and simultaneous evaluation of radiologic etiologies, including CN status, inner ear malformations, and brain abnormalities [22,23]. Such effectiveness is not possible by TBCT.

For children with prelingual congenital SSD, a thorough evaluation of CN integrity via imaging modalities within the critical period is vital because imaging studies provide information to plan for treatment strategies adequately [8,9]. Despite subtle impairment of CN, it could typically hinder the outcomes and usage of CI. TBCT has been widely used as the first-line imaging modality to evaluate all types of SNHL [4,24], which provide high-resolution images of bony canals passing CN, such as IAC and BCNC. Several studies reported that the diameter of BCNC in TBCT image is a reliable radiological landmark to estimate CN status [10,25], suggesting a strong predictive value for CND on the basis of narrow BCNC with 78.9% sensitivity and 100% specificity in pediatric subjects [25]. However, the radiological landmarks on TBCT images, including narrow IAC and narrow BCNC, albeit with a relatively high positive predictive value, cannot be a reliable predictor for abnormal CN status due to their low negative predictive value, especially in ‘small CN’ or ‘indeterminate CN’ cases with SSD. In accordance with our results, the previous study demonstrated that five out of 15 subjects with CND indicated normal BCNC and normal IAC [26].

### 4.2. A Proposed Classification of Cochlear Nerve Deficiency in This Study

Several previous studies have reported that CND is usually defined via direct comparison of the size of CN and adjacent FN on an oblique parasagittal view of MR images [27,28]. Although there is no disagreement in defining ‘absent CN’, there is no clear consensus on the definitions of ‘small CN’ as well as ‘indeterminate CN’ in the literature. We proposed the following to push forward a debate on defining both ‘small CN’ and ‘indeterminate CN’. First, CN diameter was larger, or, at equal in size to FN, in only 64% of unaffected ears, based on a direct comparison by eye measurement [29]. Therefore, a substantial portion of cases can be misdiagnosed solely based on a direct size comparison, via eye measurement, between CN and FN. Additionally, the size of FN in children under the age of five years is significantly smaller than that in older children, while there is no variance in size of CN with age [30]. Thus, ‘small CN’ or ‘indeterminate CN’ may be misdiagnosed as ‘normal CN’ in those under the age of five years. Contrastingly, our present study defined ‘small CN’ and ‘indeterminate CN’ by comparing the measurement values of Ld, Sd, and CSA of CN in the affected ear with the baseline references (the measurement values of unaffected ear) measured at the midpoint of IAC on an oblique parasagittal image acquired from 3T MRI. However, our novel definition of CND has some limitations that should be addressed in future studies. In this study, there is no investigation of the stability of CN in size concerning the patients’ age over time. Our interpretation may also be limited due to small sample size; however, the mean CN diameter of the normal ear (Ld: 1.19 ± 0.13 mm, Sd: 0.95 ± 0.16 mm) found in this study was similar to that found in a previous histopathological study (1.04 ± 0.11 mm) [31]. A future study with a larger sample size will be warranted to address the issues of etiological relevance of ‘small CN’ and ‘indeterminate CN’ in pediatric SSD.

### 4.3. Clinical Implications of ‘Indeterminate CN’

Although the CI outcomes in children with congenital SSD has not yet clearly elucidated, implantation within the critical period appeared to elicit some of the benefits of binaural integration [7,32]. Recently, CI in SSD pediatric patients with proven intact CN has been implicated regarding significant audiological and subjective benefits, even after critical period for brain plasticity [33]. On the other hand, children with CND demonstrated reduced response to electrical stimuli compared with those with intact CN, implying abnormalities of CN on MRI are associated with poor auditory outcomes in such patients [34,35]. A recent case-series study showed that the benefits are restricted by speech intelligibility or perception ability during a follow-up of at least one year after surgery in all children with CND, with a caution of its limited effectiveness and uncertain cost-benefit [36]. Notably, Arndt et al. presented that SSD children with CND were not compatible with CI candidacy in light of the anticipated poor outcomes which in turn may negatively impact bilateral hearing compared to the unaided condition [7]. Considering that an intact CN is required for successful hearing rehabilitation after CI surgery in children with congenital SSD [37], the integrity of the CN could be one of the predisposing factors for continued use of CI.

The characteristics underlying CND in SSD have previously been described [10]; however, no study has discussed the ‘indeterminate CN’ in detail to date. Here, we suggested a putative clinical implication of ‘indeterminate CN’ as a novel radiologic etiology related to SSD. It is possible that ‘indeterminate CN" in the four subjects from our original and supplementary cohort would have been undetected even by MRI if we just optically compared the size of the FN and CN by eye measurement. A recent study that investigated the outcomes of postoperative categories of auditory perception (CAP) scores in deaf subjects who underwent CT via the eye measurement method of just optically comparing the size of FN and CN reported that subjects with smaller CN than FN showed a CAP score of ≤3 after CI, while those with larger CN than FN showed a postoperative CAP score of 2 to 6 [38]. Taking this together with our results, the optical comparison of size between CN and FN via the eye measurement method is not sensitive enough to detect subtle CND, which will likely result in considering non-normal CN to be regarded as normal. Our criteria of using a comparison to the baseline size of the unaffected ear in conjunction with the optical comparison of the size between CN and FN via eye measurement seemed to provide a superior outcome than just an optical comparison via eye measurement in detecting CND. Based on a recently proposed hypothesis on the pathophysiology of CND regarding the alteration of the embryological development of CN [39], ‘indeterminate CN’ can be positioned somewhere along the CND phenotypic spectrum. Thus, we showed that ‘indeterminate CN’ was detected more frequently in ears with SSD than in those without SSD, suggesting there might be a causal relationship between ‘indeterminate CN’ and SSD. However, it seems presumptive to suggest bringing nearly significant results into clinical practice; hence, additional data will be warranted to ensure the implication of ‘indeterminate CN’ is robust. Thus, we carefully suggest extending the criteria of CND to include ‘indeterminate CN’ in children under the age of five years. 

It is worth noting that a misinterpretation of the CN status may lead to inappropriate therapeutic decisions or unsatisfied treatment outcome [40]. In this regard, the meticulous evaluation of CN proposed in our study play a pivotal role in the determination of auditory rehabilitation strategy or patients counseling. Although this hindering effect of ‘indeterminate CN’ on CI has never been investigated, more conservative attitudes toward the selection of candidates with normal CN status are necessary to avoid any attenuation of functional outcome after CI in those with SSD. The rationale for performing IAC MRI in prelingual SSD cases as the first screening, while applying our criteria for CND diagnosis, should be highlighted for subjects under the age of five years with SSD. Our rationale can be more potentiated, given the radiation-related hazard of TBCT to children.

## 5. Conclusions

IAC MRI appears to be clearly advantageous over TBCT in identifying CND in pediatric subjects with congenital SSD in terms of evaluation of the CN status which is prerequisite for confirmation of CI candidacy. Specifically, the advantage includes a capability to detect ‘small CN’ or ‘indeterminate CN’, which would not have been possible solely by TBCT. The ‘indeterminate CN’ is a novel radiological abnormality entity newly brought up in our present study. A causal relationship of the ‘indeterminate CN’ and pediatric patients with congenital SSD seems to be possible but awaits further confirmation.

## Figures and Tables

**Figure 1 jcm-07-00515-f001:**
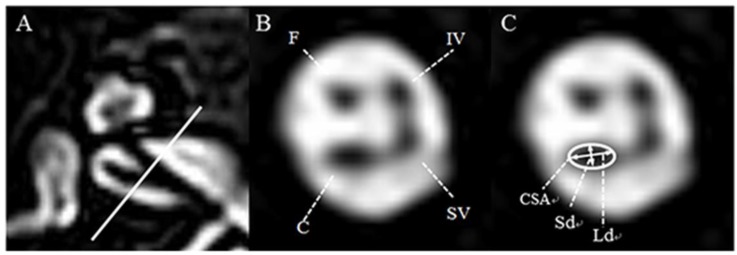
Axial and oblique sagittal three-dimensional (3D) T2-VISTA MRI of the normal ear at the level of the internal auditory canal. (**A**) The oblique sagittal image was established with the baseline perpendicular to the course of the nerve in the midpoint of internal auditory canal (IAC) of an axial image. (**B**) In the anterior aspect of the IAC, the facial nerve lies superiorly, whereas the cochlear nerve lies inferiorly. In the posterior aspect of the IAC, the superior vestibular nerve lies superiorly, whereas the inferior vestibular nerve lies inferiorly. (**C**) For the evaluation of cochlear nerve integrity, the short diameter (Sd, mm), long diameter (Ld, mm), and cross-sectional area (CSA, mm^2^) were measured as indicated. F: facial neve, C: cochlear nerve, SV: superior vestibular nerve, IV: inferior vestibular nerve.

**Figure 2 jcm-07-00515-f002:**
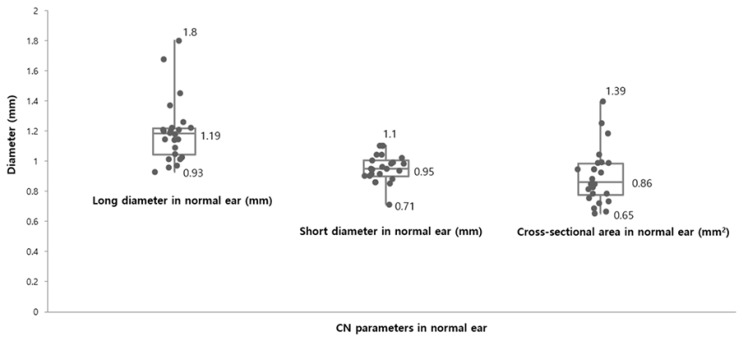
Cochlear nerve size parameters in the unaffected ears. The mean of long diameter (Ld, mm) and short diameter (Sd, mm) were 1.19 ± 0.13 mm (range: 0.93 mm–1.8 mm) and 0.95 ± 0.16 mm (range: 0.71 mm–1.1 mm), respectively. The cross-sectional area (CSA, mm^2)^ in normal ears ranged from 0.65 mm^2^ to 1.39 mm^2^ and the mean of the CSA were measured 0.86 ± 0.18 mm^2^ as present.

**Figure 3 jcm-07-00515-f003:**
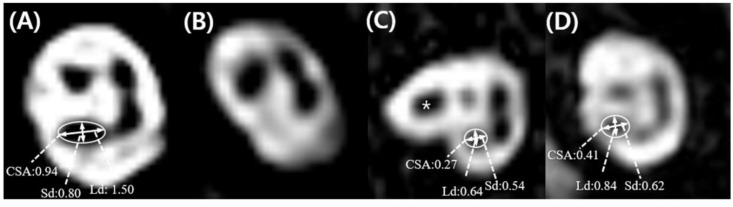
Parasagittal oblique images of the single-sided deafness ears, and the mean values of cochlear nerve size parameters according to cochlear nerve status. (**A**) Subject 2, ‘normal cochlear nerve (CN)’, where the CN is the optically larger than the facial nerve (FN). (**B**) Subject 9, ‘absent CN’ with no visible CN. (**C**) Subject 7, ‘small CN’ where the CN of the affected ear is optically smaller than the FN (asterisk) and the values of the long diameter (Ld), short diameter (Sd), and cross-sectional area (CSA) were all less than those corresponding to one standard deviation from the mean of normal ear (<1.06 mm for Ld; <0.79 mm for Sd; <0.71 mm^2^ for CSA). In this figure, the continuation of the FN located anterosuperiorly in the midpoint of the IAC was observed (**D**) Subject 5, ‘indeterminate CN’, where the CN is optically similar to the FN in size, but meets all criteria values for a ‘small CN’.

**Table 1 jcm-07-00515-t001:** Inter-rater reliability in the measurement of mean values of cochlear nerve size in the normal ears and criteria of defining CND.

	Ld (mm)	Sd (mm)	CSA (mm²)
Observer A	1.2 ± 0.17	0.96 ± 0.14	0.91 ± 0.19
Observer B	1.19 ± 0.09	0.94 ± 0.17	0.87 ± 0.17
Mean values	1.19 ± 0.13 (IRR: 0.982 *)	0.95 ± 0.16 (IRR: 0.968 *)	0.89 ± 0.18 (IRR: 0.932 *)
Criteria of defining CND	<1.06	<0.79	<0.71

Data are presented as mean ± standard deviation for numeric variables and nominal variables; Ld, long diameter; Sd, short diameter; CSA, cross sectional area; IRR, inter-rater reliability; CND, cochlear nerve deficiency; *: *p* < 0.05, values were calculated with use of *Kappa* statistic for Inter-rater reliability.

**Table 2 jcm-07-00515-t002:** Clinical values of the diameters of IAC and BCNC for estimating CND.

	IAC Diameter		BCNC Diameter	
	Narrow (<3 mm)	Normal (≥3 mm)	Narrow (<1.4 mm)	Normal (≥1.4 mm)
CND	8	11	15	4
No CND	2	3	0	5
Total	10	14	15	9
	**Narrow IAC (Value, %)**		**Narrow BCNC (Value, %)**	
Sensitivity	42.1%		78.9%	
Specificity	60.0%		100.0%	
Accuracy	45.8%		83.3%	
Positive predictive value	80.0%		100.0%	
Negative predictive value	21.4%		55.6%	

CND, cochlear nerve deficiency; IAC, internal acoustic canal; BCNC, bony cochlear nerve canal.

**Table 3 jcm-07-00515-t003:** CND identification rate via the estimation from the presence of narrow IAC and narrow BCNC as well as FN-CN size optical comparison in combination with baseline reference.

	IAC Diameter (TBCT)		BCNC Diameter (TBCT)		Parasagittal Oblique Images (MRI)	
	Normal (≥3 mm)	Narrow (<3 mm)	Normal (≥1.4 mm)	Narrow (<1.4 mm)	FN < CN in Size (Normal CN)	FN ≥ CN in Size (CND)
Normal CN	3	2	5	0	5	0
Absent CN	7	8	0	15	0	15
Small CN	2	0	2	0	0	2
Indeterminate CN	2	0	2	0	0	2
	**Narrow IAC** **(based on TBCT)**		**Narrow BCNC** **(based on TBCT)**		**FN ≥ CN in Size** **(based on MRI)**	
CND identification rate (%)	8/24 (33.3)		15/24 (62.5)		19/24 (79.2) *	

CN, cochlear nerve; CND, cochlear nerve deficiency; TBCT, temporal bone computed tomography; IAC, internal acoustic canal; BCNC, bony cochlear nerve canal; FN, facial nerve; CN, cochlear nerve. *: *p* < 0.05, comparisons were calculated by means of Wilcoxon signed-rank test for CND identification rate between two imaging modalities. (Narrow IAC vs. FN ≥ CN in size, *p* = 0.003, Narrow BCNC vs. FN ≥ CN in size, *p* = 0.0456).

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
