# Peer review of "Appropriate Imaging Modality for the Etiologic Diagnosis of Congenital Single-Sided Deafness in Children"

_jcm, 2018, doi:10.3390/jcm7120515_

Round 1
Reviewer 1 Report
The paper describes an interesting new clinical implication of indeterminate CN as a novel radiologic etiology related to SSD, but the description of MRI measurement is too long and detailed and doesn't add any new information.
Minor revision:
In material and methods, the paragraph: "size measurement of inner
ear in IAC MRI" is too long an too much technical. It must be rewritten
in a simpler way to make it accessible even to non-radiologists.
Author Response
First of all, we appreciate all academic editor and two reviewers’ precious comments and suggestions. All the changes made in the manuscript has been written in blue, and the MS word track-change function has been activated for the reviewers’ convenience.
Reviewer #1.
The paper describes an interesting new clinical implication of indeterminate CN as a novel radiologic etiology related to SSD, but the description of MRI measurement is too long and detailed and doesn't add any new information. Minor revision: In material and methods, the paragraph: "size measurement of inner ear in IAC MRI" is too long an too much technical. It must be rewritten in a simpler way to make it accessible even to non-radiologists.
[Response] Thank you for pointing out a wordy description.
In response to reviewer’s suggestion, we have shorten up the description significantly. Now it reads
“IAC imaging was performed by 3T MRI scanner (Achieva and Ingenia; Philips, Best, Netherlands), equipped with SENSE head coil (Philips Healthcare) for signal reception. On the basis of T2-weighted turbo spin-echo acquisition (TR/TE: 7395/120ms; flip angle: 90˚; slice thickness: 0.7 mm; no slice gap; FOV: 100 × 100 mm2; voxel size: 0.4 × 0.4 × 0.4 mm3) The oblique sagittal images that directly show the presence of nerves in the IAC were reformatted in planes perpendicular to the course of the nerve in the midpoint of IAC of axial images (Fig. 1A). On the oblique parasagittal image of the unaffected ear, the facial nerve (FN) lies superiorly, with the cochlear nerve (CN) inferior to it, within the anterior aspect of the canal. Superior (SVN) and inferior vestibular nerve (IVN) lie posteriorly (Fig. 1B). The lengths of the major axis (long diameter, Ld) and the minor axis, which is perpendicular to the major axis (short diameter, Sd) of CN, on the oblique parasagittal images of both the unaffected and affected ears were measured to the nearest 0.1 mm by two neuro-radiologists blinded to all subjects’ information using a software on the Picture Archiving and Communication System (PACS) workstation combined with sharpen and zooming (Infinitt, Seoul, South Korea). The cross-sectional area (CSA) of CN was also measured as elliptical regions of interest (ROI) on the basis of Sd and Ld (Fig. 1C).
The mean of Ld, Sd, and CSA in the unaffected ears from the present study were 1.19±0.13mm, 0.95±0.16mm, and 0.89±0.18mm2, respectively (Fig.2) The CN status was classified in accordance with the presence or the size of CN: (1) ‘absent CN’ refers to the a parasagittal oblique image with no visible CN; (2) ‘small CN’ is defined as when CN of the affected ear is optically smaller than FN and the values of Ld, Sd, and CSA were all less than those corresponding to one standard deviation from the mean of the unaffected ear (Ld<1.06mm ; Sd <0.79mm; CSA<0.71mm2); (3) ‘indeterminate CN’ is defined as when CN is optically similar to FN in size but meets all diagnostic criteria values for ‘small CN'. In this study, CND includes ‘absent CN’, ’small CN’, and ‘indeterminate CN’ as defined according to their criteria described above. Based on IAC MRI, anatomical abnormalities were reviewed with regard to structural specificities, including CN integrity, inner ear (cochlea, vestibule, endolymphatic duct, semicircular canals), and brain lesions.”

Reviewer 2 Report
This is an interesting idea, but the title and focus of the paper are misgiuded. There is already a much larger study on the role of CT and MRI in congenital deafness:
https://www.ncbi.nlm.nih.gov/pubmed/27891421
This paper concluded that both imaging modalities are needed.
The current paper adds little to that debate, and the conclusion that MRI is sufficient seems to be based on the flawed assumption that most congenital deafness is cause by cochlear nerve dyspalsia, and a much smaller sample size than that study above.
I think there is value in this paper - in particular looking at defining imaging characteristics of the cochlear nerve on MRI, and in particular relating that in some way to patient outcomes (the latter are not mentioned at all). However, that topic may be too specialised for this journal.
Author Response
Reviewer #2.
This is an interesting idea, but the title and focus of the paper are misguided. There is already a much larger study on the role of CT and MRI in congenital deafness: https://www.ncbi.nlm.nih.gov/pubmed/27891421. This paper concluded that both imaging modalities are needed. The current paper adds little to that debate, and the conclusion that MRI is sufficient seems to be based on the flawed assumption that most congenital deafness is cause by cochlear nerve dysplasia, and a much smaller sample size than that study above. I think there is value in this paper - in particular looking at defining imaging characteristics of the cochlear nerve on MRI, and in particular relating that in some way to patient outcomes (the latter are not mentioned at all). However, that topic may be too specialized for this journal.
[Response] Thank you very much for great comments.
Please refer to our response to the Editor’s comment.
In this study, MRI appeared to be advantageous over TBCT in a meticulous evaluation of CN which is crucial for selection of appropriate CI candidacy among congenital SSD children. Furthermore, given that a significant higher incidence of cochlear nerve deficiency (CND) in patients with SSD than those with BD, our novel methodology should be a perfect fit in evaluating those patients. Further, we, for the first time, suggest a novel concept of ‘indeterminate CN’, of which causal relationship with SSD awaits confirmation. We believe this study will be widely read and in various fields of medicine such as otolaryngology, audiology, and neurosurgery
In response to reviewer’s comment, we have added and revised the discussion section with reflection the opinions of the reviewers.

Round 2
Reviewer 2 Report
I appreciate the authors opinions, but without any reported outcomes of cochlear implantation, this paper simply describes a way to classify the cochlear nerve but provides no useful information on how this can be used in clinical decision making. It will in no way change my practice.
Author Response
RESPONSE TO THE REVIEWER’S COMMENTS:
First of all, we appreciate the reviewers’ precious comments and suggestions. All the changes made in the manuscript has been written in blue, and the MS word track-change function has been activated for the reviewers’ convenience.
Reviewer #2.
I appreciate the authors opinions, but without any reported outcomes of cochlear implantation, this paper simply describes a way to classify the cochlear nerve but provides no useful information on how this can be used in clinical decision making. It will in no way change my practice.
[Response] Thank you for providing us an opportunity for clarification.
Although the outcomes of cochlear implant (CI) in children with congenital single-sided deafness (SSD) has not yet clearly elucidated, implantation within the critical period appeared to elicit some of the benefits of binaural integration [1,2]. Recently, CI in SSD pediatric patients with proven intact cochlear nerve (CN) has been implicated regarding significant audiological and subjective benefits, even after critical period for brain plasticity [3]. On the other hand, children with cochlear nerve deficiency (CND) demonstrated reduced response to electrical stimuli compared with those with intact CN, implying abnormalities of CN on MRI (magnetic resonance imaging) are associated with poor auditory outcomes in such patients[4,5]. A recent case-series study showed that the benefits are restricted by speech intelligibility or perception ability during a follow-up of at least one year after surgery in all children with CND, with a caution of its limited effectiveness and uncertain cost-benefit [6]. Notably, Arndt et al. presented that SSD children with CND were not compatible with CI candidacy in light of the anticipated poor outcomes which in turn may negatively impact bilateral hearing compared to the unaided condition [7]. Considering that intact CN is required for successful hearing rehabilitation after CI surgery in children with congenital SSD [8], the integrity of CN could be one of the predisposing factors for continued use of CI.
In this perspective, we suggested the clinical implication of cochlear nerve (CN) evaluation with special attention to ‘indeterminate CN’. This study also presented the validity for CN measurement we proposed, not the existing method (comparison of FN and CN) in discussion. Given that misinterpretation of the CN status may elicit inappropriate therapeutic decisions or unsatisfied treatment outcome [9], the meticulous evaluation of CN proposed in our study play a pivotal role in the determination of auditory rehabilitation strategy or patients counseling. Collectively, more conservative attitudes toward the selection of candidates with normal CN status are necessary to avoid any attenuation of functional outcome after CI in those with SSD. The rationale for performing IAC MRI in prelingual SSD cases as the first screening, while applying our criteria for CND diagnosis, should be highlighted. Our rationale can be more potentiated, given the radiation-related hazard of CT to children. In response to the reviewer’s comment, we have added a clinical significance of our study in the discussion section.
References
1. Tavora-Vieira, D.; Rajan, G.P. Cochlear Implantation in Children with Congenital and Noncongenital Unilateral Deafness. Otol Neurotol 2015, 36, 1457-1458, doi:10.1097/MAO.0000000000000806.
2. Tavora-Vieira, D.; Rajan, G.P. Cochlear implantation in children with congenital unilateral deafness: Mid-term follow-up outcomes. Eur Ann Otorhinolaryngol Head Neck Dis 2016, 133 Suppl 1, S12-14, doi:10.1016/j.anorl.2016.04.016.
3. Thomas, J.P.; Neumann, K.; Dazert, S.; Voelter, C. Cochlear Implantation in Children With Congenital Single-Sided Deafness. Otol Neurotol 2017, 38, 496-503, doi:10.1097/MAO.0000000000001343.
4. He, S.; Shahsavarani, B.S.; McFayden, T.C.; Wang, H.; Gill, K.E.; Xu, L.; Chao, X.; Luo, J.; Wang, R.; He, N. Responsiveness of the Electrically Stimulated Cochlear Nerve in Children With Cochlear Nerve Deficiency. Ear Hear 2018, 39, 238-250, doi:10.1097/AUD.0000000000000467.
5. Peng, K.A.; Kuan, E.C.; Hagan, S.; Wilkinson, E.P.; Miller, M.E. Cochlear Nerve Aplasia and Hypoplasia: Predictors of Cochlear Implant Success. Otolaryngol Head Neck Surg 2017, 157, 392-400, doi:10.1177/0194599817718798.
6. Zhang, Z.; Li, Y.; Hu, L.; Wang, Z.; Huang, Q.; Wu, H. Cochlear implantation in children with cochlear nerve deficiency: a report of nine cases. Int J Pediatr Otorhinolaryngol 2012, 76, 1188-1195, doi:10.1016/j.ijporl.2012.05.003.
7. Arndt, S.; Prosse, S.; Laszig, R.; Wesarg, T.; Aschendorff, A.; Hassepass, F. Cochlear implantation in children with single-sided deafness: does aetiology and duration of deafness matter? Audiol Neurootol 2015, 20 Suppl 1, 21-30, doi:10.1159/000380744.
8. Friedmann, D.R.; Ahmed, O.H.; McMenomey, S.O.; Shapiro, W.H.; Waltzman, S.B.; Roland, J.T., Jr. Single-sided Deafness Cochlear Implantation: Candidacy, Evaluation, and Outcomes in Children and Adults. Otol Neurotol 2016, 37, e154-160, doi:10.1097/MAO.0000000000000951.
9. Valero, J.; Blaser, S.; Papsin, B.C.; James, A.L.; Gordon, K.A. Electrophysiologic and behavioral outcomes of cochlear implantation in children with auditory nerve hypoplasia. Ear and hearing 2012, 33, 3-18.
